# Heme oxygenase-1 as an important predictor of the severity of COVID-19

Yu Hara[1☉], Jun Tsukiji[2☉]*, Aya Yabe[1], Yoshika Onishi[3], Haruka Hirose[3], Masaki Yamamoto[4], Makoto Kudo[4], Takeshi Kaneko[1], Toshiaki Ebina[3]

1 Department of Pulmonology, Yokohama City University Graduate School of Medicine, Yokohama, Japan,
2 Department of Prevention and Infection Control, Kanagawa Cancer Center, Yokohama, Japan,
3 Department of Laboratory Medicine and Clinical Investigation, Yokohama City University Medical Center, Yokohama, Japan, 4 Respiratory Disease Center, Yokohama City University Medical Center, Yokohama, Japan

☉ These authors contributed equally to this work.
* j-tsukiji@kcch.jp

**Data Availability Statement:** All relevant data are within the paper and its Supporting information files.

## Abstract

### Background and objective

A cytokine storm is caused by inflammatory cells, including pro-inflammatory macrophage phenotype (M1), and play a critical role in the pathogenesis of COVID-19, in which diffuse alveolar damage occurs in the lungs due to oxidative stress exposure. Heme oxygenase (HO)-1 is a stress-induced protein produced by the anti-inflammatory / anti-oxidative macrophage phenotype (M2), which also produces soluble CD163 (sCD163). In our study, we investigated and determined that serum HO-1 can be a predictive biomarker for assessing both the severity and the outcome of COVID-19 patients.

### Method

The serum concentrations of HO-1 and sCD163 of COVID-19 patients were measured on admission. The relationship between these biomarkers and other clinical parameters and outcomes were evaluated.

### Results

Sixty-four COVID-19 patients (11 mild, 38 moderate, and 15 severe cases) were assessed. The serum HO-1 tended to increase (11.0 ng/mL vs. 24.3 ng/mL vs. 59.6 ng/mL with severity). Serum HO-1 correlated with serum lactate dehydrogenase (R = 0.422), C-reactive protein (R = 0.463), and the ground glass opacity (GGO) and consolidation score (R = 0.625) of chest computed tomography. The serum HO-1 showed a better area under the curve (AUC) for predicting ICU admission than the serum sCD163 (HO-1; 0.816 and sCD163; 0.743). In addition, composite parameters including serum HO-1 and the GGO and consolidation score showed a higher AUC for predicting ICU admission than the AUC of a single parameter.

**Funding:** The funders had no role in study design, data collection and analysis, decision to publish, or preparation of the manuscript.

**Competing interests:** The authors have declared that no competing interests exist.

**Abbreviations:** AE, acute exacerbation; ARDS, acute respiratory distress syndrome; AUC, area under the ROC curve; CO, carbon monoxide; COVID-19, coronavirus disease 2019; CRP, C-reactive protein; CT, computed tomography; DAD, diffuse alveolar damage; ECMO, extracorporeal membrane oxygenation; ELISA, enzyme-linked immunosorbent assay; GGO, ground glass opacity; HO, heme oxygenase; HRCT, high-resolution CT; ICU, intensive care unit; ILD, interstitial lung disease; KL-6, Krebs von den Lungen-6; LDH, lactate dehydrogenase; $PaO_2$, arterial partial oxygen pressure; RI, reference interval; ROC, receiver operating characteristic; SARS-CoV-2, severe acute respiratory syndrome coronavirus 2; sCD163, soluble CD163.

## Conclusion

Clinically, serum HO-1, reflecting the activation of M2, could be a very useful marker for evaluating disease severity and predicting prognoses for COVID-19 patients. In addition, controlling activated M2 might be a preventative COVID-19 therapeutic target.

## Introduction

The coronavirus disease 2019 (COVID-19) pandemic caused by severe acute respiratory syndrome coronavirus 2 (SARS-CoV-2), and is continuing to threaten the world. The clinical course of COVID-19 varies. Although the majority of COVID-19 patients have self-limiting forms of the disease, some patients deteriorate rapidly [1–8]. Accurate assessment of disease severity is essential when making clinical decisions regarding treatment options in order to improve the prognosis of COVID-19 patients.

Macrophages are critical contributors to immune responses in COVID-19 [9, 10]. The condition by excess production and release of inflammatory cytokines is known as a cytokine storm, in which a proliferation and activation of macrophages are driven, and this is thought to be a major cause of disease progression in COVID-19 patents.

Airborne SARS-CoV-2 particulates bind with the angiotensin-converting enzyme 2-expressing type II pneumocytes, and triggers immune-response-induced oxidative stress, which leads to lung damage [11, 12]. Type II pneumocytes reside in the alveoli wall and alveolar macrophages in its sacs, which occupy more than 90% of the resident immune cells [13]. Macrophages are broadly divisible in to two groups: pro-inflammatory phenotype (M1) and anti-inflammatory, anti-oxidative and cytoprotective phenotype (M2) [14]. It was recently clarified that SARS-CoV-2 distinctively infects M1, but not M2 [15]. The polarization of macrophage is a critical factor for the pathogenesis of COVID-19.

The surface molecules of M2 can sustain its properties. Particularly, efficient extracellular hemoglobin (Hb) clearance is critical in preventing oxidative toxicity. CD163, a scavenger receptor of a Hb-haptoglobin complex, is a macrophage-restricted transmembrane protein which is expressed on M2. Continuous shedding by proteolytic cleavage of the extracellular portion of CD163 from the surface of M2 increases the amounts of soluble receptors sCD163 [16–18], has been shown to be a valuable diagnostic marker of systemic inflammatory disorders, such that sCD163 is a disease severity marker in COVID-19 pneumonia [19, 20].

Heme oxygenase (HO) is also considered a potent effector molecule associated with M2 [21]. The mammalian heme oxygenase (HO) enzyme catalyzes the $O_2$-dependent degradation of heme into carbon monoxide (CO), iron, and biliverdin through numerous reaction intermediates [22]. Two isoforms of HO have been characterized, an inducible HO-1 and a constitutive HO-2. HO-1 is strongly and exclusively induced in M2, but not in M1 by heat shock and oxidative stress conditions. The anti-inflammatory and anti-oxidative actions of each product originating from HO-1, CO and biliverdin, sustain the properties of M2 [14, 23]. Thus, these actions and findings of HO-1 maintain homeostasis and may allow us to formulate the hypothesis that the HO-1 from M2 plays a crucial role in COVID-19. As a matter of fact, the HO-1 gene expression, *HMOX1*, has been reported in COVID-19 [24].

HO-1 is increased in alveolar macrophages in various pulmonary diseases such as acute respiratory distress syndrome (ARDS) and interstitial lung disease (ILD), reflecting the activation of M2 against oxidative stress [25, 26]. Elevated HO-1 levels in serum have been reported to be useful in evaluating the severity of lung damage in ILD and for predicting subsequent fibrosis formation [27–29].

In this study, we investigated whether or not serum HO-1, which reflects the degree of oxidative stress in COVID-19 patients, could be a useful biomarker in predicting disease severity and determining a prognosis by making a comparison to those of sCD163.

## Methods

### Study location and patients

This was a single-center retrospective study at the Yokohama City University Medical Center in Yokohama, Japan. Our study includes inpatients and outpatients with confirmed COVID-19, who were admitted to our facility between February, 2020 and December, 2020. Our data included age, sex, smoking history, comorbidities, laboratory data, high-resolution CT (HRCT) findings, treatments, and outcomes, including intensive care unit (ICU) admissions, and mechanical ventilation or extracorporeal membrane oxygenation (ECMO) support. The HRCT findings were assessed independently by a pulmonologist and a radiologist and were evaluated using the semi-quantitative scoring method described by Ooi *et al.* [30]. Abnormalities on HRCT images of lungs were categorized as ground glass opacity (GGO), consolidation, and reticular fibrosis including honeycomb (S1 Fig), and were scored based on ratios (%) of disease in each of the six lung lobes (0%: 0 point, 1–25%: 1 point, 26–50%: 2 points, 51–75%: 3 points, 76%-: 4 points). Global scores were calculated by adding the scores for each abnormality in all lobes and these scores were expressed as the GGO and consolidation score or reticular fibrosis score. In addition, cases in which HRCT could be re-examined, scoring was performed again.

### Diagnosis and severity classification of COVID-19

SARS-CoV-2 RNA was detected using real-time reverse transcription quantitative polymerase chain reaction (PCR) on nucleic acids extracted from nasopharyngeal and pharyngeal swab samples from all patients upon admission or upon arrival as an outpatient. In Japan, the disease severity of COVID-19 is generally grouped in 1 of 4 categories, based on the Japanese Ministry of Health Labor and Welfare guidelines [31]. Mild cases were defined as having slight symptoms without pneumonia on HRCT. Moderate grade I cases were defined as having mild respiratory symptoms with pneumonia and slight hypoxemia ($93\% < SpO_2 < 96\%$), but without oxygen supplementation. Moderate grade II cases were defined as having $SpO_2 \leq 93\%$ requiring oxygen supplementation. Severe cases were defined as requiring mechanical ventilation or ECMO support. In this study, mild and moderate grade I cases were combined into "mild cases," because the clinical features of the mild and moderate type I cases were similar, and the number of mild cases was very small. The moderate grade II cases were defined as "moderate cases."

### Blood samples and details of serum HO-1 and sCD163

Blood samples from all patients were taken upon admission or upon arrival as an outpatient. The following were measured; white blood cell count (reference interval (RI): 3300–8600 /μL), platelet count (RI: $15.8\times10^4$–$34.8\times10^4$ /μL), total bilirubin (T-bil; RI: 0.4–1.5 mg/dL), albumin (RI: 4.1–5.1 g/dL), creatinine (RI: 0.65–1.07 mg/dL (male), 0.46–0.79 mg/dL (female)), LDH (RI: 124–222 U/L), CRP (RI: 0.000–0.140 mg/dL), and Krebs von den Lungen-6 (KL-6; RI: <500 U/mL). Serum HO-1 levels were measured on admission using the ImmunoSet™ HO-1 enzyme-linked immunosorbent assay (ELISA) development set (Enzo, Farmingdale, NY, USA) and sCD163 levels were measured using Quantikine ELISA Human CD163 Immunoassay (R&D Systems, Minneapolis, MN, USA) according to the manufacturers' instructions. The

details of this ELISA method for serum HO-1 measurement were described previously [27, 28]. The serum concentrations of HO-1 and sCD163 of normal controls were measured from 19 biobank samples from healthy volunteers before 2019.

## Statistical analysis

Data are shown as medians with the $25^{th}$–$75^{th}$ percentiles or numbers (%). Statistical analysis was performed using JMP11 (SAS Institute, Inc., Cary, NC, USA). Group comparisons were made using Wilcoxon's rank-sum test, the chi-squared test, or paired t-test as appropriate. Spearman's correlation coefficients were calculated to assess the relationship between serum HO-1 and sCD163 and other clinical parameters. The utility of serum HO-1 with or without other clinical parameters for predicting outcomes was evaluated by area under the receiver operating characteristic (ROC) curve (AUC) analysis. P values < 0.05 were considered significant.

## Study approval

This study was performed in accordance with the Declaration of Helsinki and approved by the institutional review board of the Yokohama City University (approval number B200500023). In all patients, consent for participation in our retrospective study was obtained by disclosing this as a clinical study, and we included an opt-out option (https://yokohama-cu.bvits.com/rinri/publish.aspx).

# Results

## Patient characteristics

Table 1 shows the clinical characteristics of the patients with COVID-19, including 11 patients with mild disease, 38 with moderate disease, and 15 with severe disease. Seven were outpatient and 57 patients were admitted. The incidence of men and diabetes mellitus tended to increase according to disease severity. In the severe group, many patients were admitted to the ICU and needed mechanical ventilation and ECMO support, however, even in the moderate group, 9 patients (24%) were admitted to the ICU and 4 (11%) received mechanical ventilation support and 2 (5%) received ECMO support. Six patients died due to respiratory failure in our hospital but no postmortems analyses were performed. Both serum HO-1 and sCD163 levels were higher in COVID-19 patients than in healthy volunteers (median HO-1: 27.5 ng/mL vs. 2.9 ng/mL (P < 0.001), median sCD163 615.0 ng/mL vs. 368.5 ng/mL (P < 0.001)). Similar with serum LDH, CRP, and creatinine, these levels increased according to disease severity (Fig 1A and 1B). In addition, comparing the prediction of the disease severity between serum HO-1 and sCD163 (severe vs. moderate and mild), the AUC was higher for HO-1 (0.875) than sCD163 (0.733) (Fig 1C). No pulmonary function tests were performed in the COVID-19 cases during their admissions due to the prevention against infection we undertook in order to avoid the pathogens spread in the hospital.

## Relationship between the serum HO-1 and sCD163 and other clinical parameters

As shown in Table 2, both serum HO-1 and sCD163 levels showed a positive correlation with serum LDH levels, CRP and the GGO and consolidation scores calculated from HRCT, and a negative correlation with serum albumin. Although sample numbers were small, both serum HO-1 and sCD163 levels showed a positive correlation with the serum ferritin levels. Interestingly, with serum KL-6, only sCD163 was positively correlated (S1 Table).

**Table 1. Patients' characteristics.**

| | Mild (N = 11) | Moderate (N = 38) | Severe (N = 15) | P values |
|---|---|---|---|---|
| **Age** | 68 (33–78) | 70 (59–77) | 67 (57–73) | 0.647 |
| **Male / Female** | 2 / 9 | 24 / 14 | 11 / 4 | 0.011 |
| **Current smoker** | 2 (18) | 3 (8) | 1 (7) | 0.307 |
| **Comorbidity** | | | | |
| **Hypertension** | 6 (55) | 14 (37) | 10 (67) | 0.196 |
| **Diabetes mellitus** | 2 (18) | 12 (32) | 9 (60) | 0.146 |
| **COPD** | 1 (9) | 3 (8) | 1 (7) | 0.221 |
| **Cardiovascular disease** | 2 (18) | 12 (32) | 2 (13) | 0.327 |
| **Malignancy** | 3 (27) | 10 (26) | 1 (7) | 0.265 |
| **Blood biomarkers** | | | | |
| **WBC, μL** | 5390 (4400–7400) | 5600 (4200–7800) | 9400 (4800–15000) | 0.091 |
| **Neutrophils, /μL** | 2884 (2307–4710) | 3833 (2317–5466) | 3620 (1109–8668) | 0.592 |
| **Lymphocytes, /μL** | 1391 (540–1640) | 831 (519–1320) | 529 (285–936) | 0.046 |
| **Platelet, $10^4$ μL** | 19.3 (16.2–21.0) | 20.0 (13.8–25.4) | 19.7 (11.9–27.9) | 0.942 |
| **Total bilirubin, mg/dL** | 0.7 (0.5–0.7) | 0.5 (0.4–0.8) | 0.6 (0.5–1.1) | 0.476 |
| **LDH, U/L** | 214 (169–270) | 257 (197–318) | 430 (349–584) | < 0.001 |
| **Albumin, g/dl** | 4.0 (3.7–4.3) | 3.1 (2.7–3.8) | 2.8 (2.5–3.4) | 0.001 |
| **Creatinine, mg/dL** | 0.67 (0.59–0.88) | 0.73 (0.6–0.85) | 1.00 (0.82–1.17) | 0.016 |
| **CRP. mg/dL** | 0.76 (0.10–4.93) | 3.47 (1.13–7.99) | 10.11 (5.52–17.67) | < 0.001 |
| **HO-1, ng/mL** | 11.0 (7.3–33.4) | 24.3 (13.5–35.1) | 59.6 (41.1–100) | < 0.001 |
| **Soluble CD163, pg/mL** | 502.3 (394.6–591.5) | 631.8 (449.3–816.7) | 791.0 (597.8–1575.4) | 0.008 |
| **HRCT scores** | | | | |
| **GGO and consolidation score** | 3.5 (1–4.3) | 6 (4–10) | 13 (12–17) | < 0.001 |
| **Reticular fibrosis score** | 0 | 0 (0–2) | 0 | 0.043 |
| **Treatment** | | | | |
| **Favipiravir** | 0 (0) | 2 (5) | 2 (13) | 0.353 |
| **Ciclesonide** | 0 (0) | 5 (13) | 5 (33) | 0.056 |
| **Remdesivir** | 1 (9) | 23 (61) | 12 (80) | 0.001 |
| **Dexamethasone** | 1 (9) | 19 (50) | 10 (67) | 0.012 |
| **Lopinavir** | 0 (0) | 3 (8) | 3 (20) | 0.633 |
| **Tocilizumab** | 0 (0) | 4 (11) | 1 (7) | 0.510 |
| **ICU visit** | 0 (0) | 9 (24) | 15 (100) | < 0.001 |
| **Mechanical ventilation** | 0 (0) | 4 (11) | 14 (93) | < 0.001 |
| **ECMO** | 0 (0) | 2 (5) | 5 (33) | 0.006 |
| **Hospital mortality** | 0 (0) | 1 (3) | 5 (33) | < 0.001 |

Results are shown as medians with the 25th–the 75th percentiles or numbers (%).

**Abbreviation lists**

COPD, chronic obstructive pulmonary disease; CRP, C-reactive protein; ECMO, extracorporeal membrane oxygenation; GGO, ground glass opacity; HO, heme oxygenase; HRCT, high-resolution computed tomography; ICU, intensive care unit; LDH, lactate dehydrogenase; WBC, white cell count.

## Predicting ICU admission and mechanical ventilation or ECMO support

The ROC curve analyses of the serum HO-1 (red line) and sCD163 (blue line) levels for predicting outcomes, including ICU admission and mechanical ventilation or ECMO support, were performed. The AUCs for predicting ICU admission were 0.816 for serum HO-1 and 0.743 for sCD163 (Fig 2A). Eighteen patients (28%) required mechanical ventilation support and 7 patients (11%) ECMO support. The AUCs for predicting mechanical ventilation support

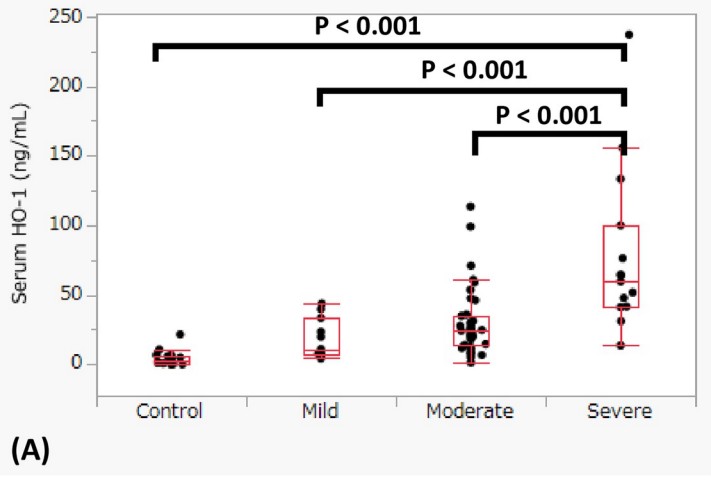

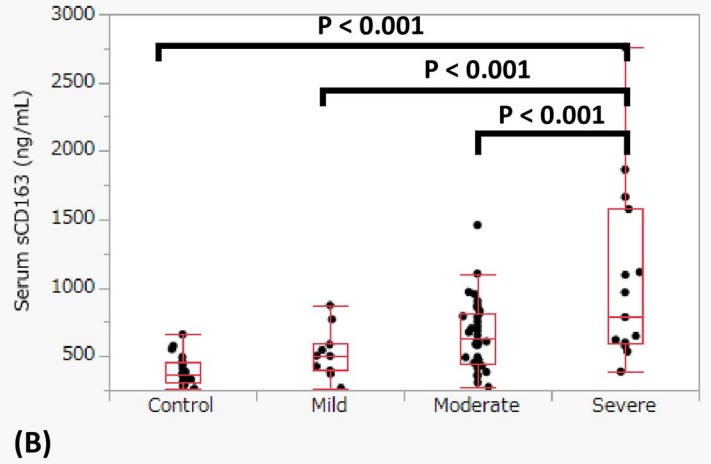

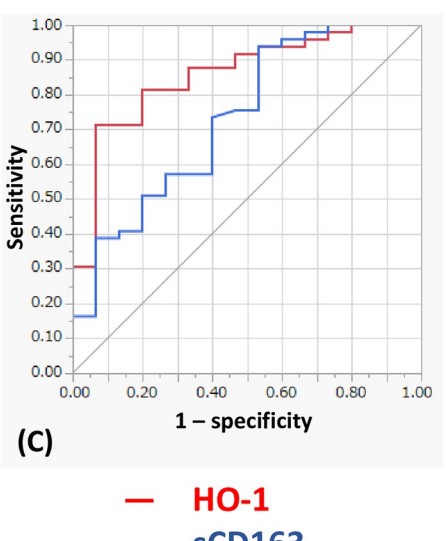

**Fig 1. Serum heme oxygenase (HO)-1 and soluble CD163 (sCD163) with COVID-19.** (A) HO-1 according to disease severity; (B) sCD163 according to disease severity; (C) Comparison of the prediction of severity between HO-1 (red line) and sCD163 (blue line). (A, B) Both serum HO-1 and sCD163 indicated high levels according to the disease severity of COVID-19. The center bold line is the median value; the bottoms and tops of the boxes represent the 25th and 75th percentiles, respectively; and the whiskers are 95% confidence intervals. (C) Comparing the prediction of disease severity between HO-1 and sCD163 (severe vs. moderate and mild), serum HO-1 shows a higher AUC (0.875) than sCD163 (0.733).

**Table 2. The relationships between serum HO-1 or sCD163 and other clinical parameters.**

| | | N | R | 95%CI | P values |
|---|---|---|---|---|---|
| Serum HO-1 | Serum LDH | 62 | 0.422 | 0.193–0.608 | < 0.001 |
| | Serum CRP | 63 | 0.463 | 0.243–0.638 | < 0.001 |
| | Serum albumin | 63 | -0.332 | -0.536–-0.092 | 0.008 |
| | GGO and consolidation score | 61 | 0.625 | 0.442–0.757 | < 0.001 |
| Serum sCD163 | Serum LDH | 62 | 0.373 | 0.136–0.570 | 0.003 |
| | Serum CRP | 63 | 0.333 | 0.093–0.537 | 0.008 |
| | Serum albumin | 63 | -0.399 | -0.588–-0.167 | 0.001 |
| | GGO and consolidation score | 61 | 0.525 | 0.315–0.686 | < 0.001 |

**Abbreviation lists**

CRP, C-reactive protein; GGO, ground glass opacity; HO, heme oxygenase; LDH, lactate dehydrogenase.

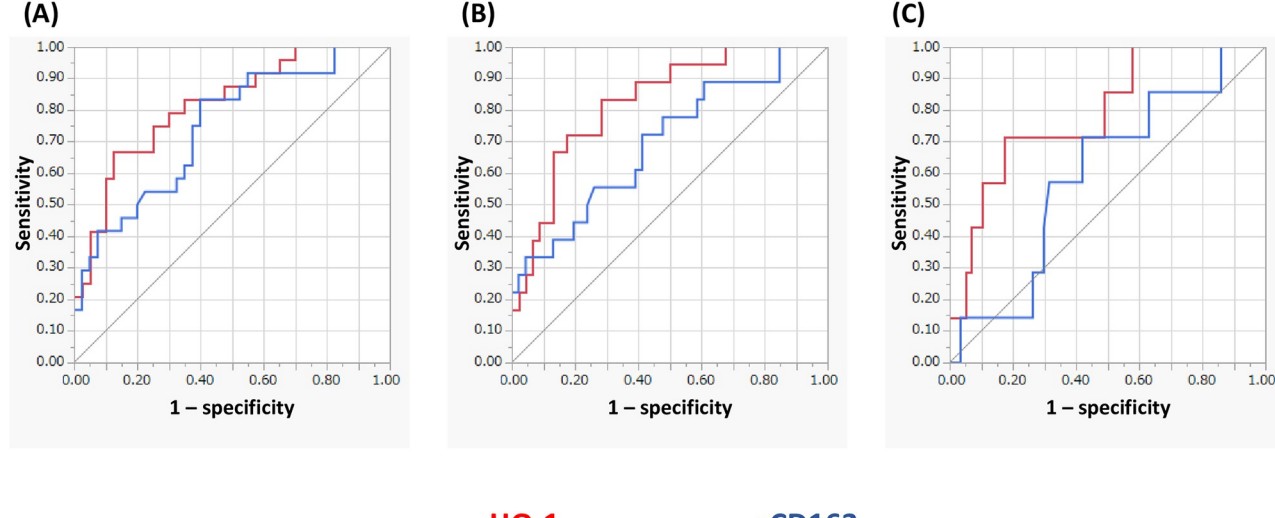

**Fig 2. Prediction of intensive care unit (ICU) admission (A), mechanical ventilation support (B), and extracorporeal membrane oxygenation (ECMO) support (C) using serum heme oxygenase (HO)-1 (red line) and soluble CD163 (sCD163) levels (blue line).** The receiver operating characteristic (ROC) curve analyses for serum HO-1 and sCD163 levels to predict treatment outcomes including ICU admission and mechanical ventilation or ECMO support. The area under the ROC curve (AUC) for predicting ICU admission was 0.816 for serum HO-1 and 0.743 for sCD163 (A). The AUC for predicting mechanical ventilation support is 0.827 for serum HO-1 and 0.696 for sCD163 (B). The AUCs for predicting ECMO support are 0.790 and 0.598, respectively (C).

were 0.827 for serum HO-1 and 0.696 for sCD163. The AUCs for predicting ECMO support were 0.790 and 0.598, respectively (Fig 2B and 2C).

## Combination of serum HO-1 and GGO and consolidation score

Predicting ICU admission using a composite parameter including serum HO-1, sex, and the GGO and consolidation score showed a higher AUC than a single predictor (only HO-1, 0.816) or a combination of the two (Fig 3A and 3C). Furthermore, composite parameters, including serum HO-1, sex, and the GGO and consolidation score for predicting mechanical ventilation support showed a higher AUC than a single predictor (only HO-1, 0.827) or a combination of the two (Fig 3B and 3D).

## Relationship between baseline serum HO-1 and the serial change of GGO and consolidation score

We followed up the HRCT scores in 31 patients (the median duration from first to second HRCT was 26 days) and divided these cases into a low baseline HO-1 group (N = 16) and a high baseline HO-1 group (N = 15). The former group consisted of 2 mild cases, 13 moderate cases, and 1 severe case (including 1 fatality (6%)) and the latter group consisted of 6 moderate cases and 9 severe cases (including 3 fatalities (20%)). We found that in the former group, the GGO and consolidation score was significantly decreased (first GGO and consolidation score, 5 (2.3–7.8 points); second GGO and consolidation score, 0.5 (0–6.8 points) (P = 0.017)) (Fig 4A), while in the latter group, it was persistently higher (first GGO and consolidation score, 13 (10–17 points); second GGO and consolidation score, 15 (8–22 points) (P = 0.828)) (Fig 4B).

**(A)**

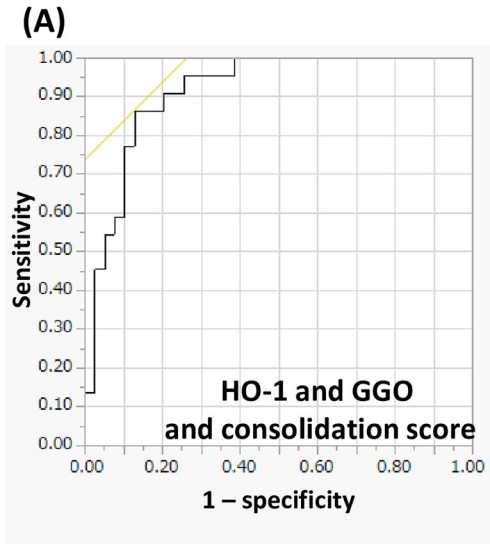

**(B)**

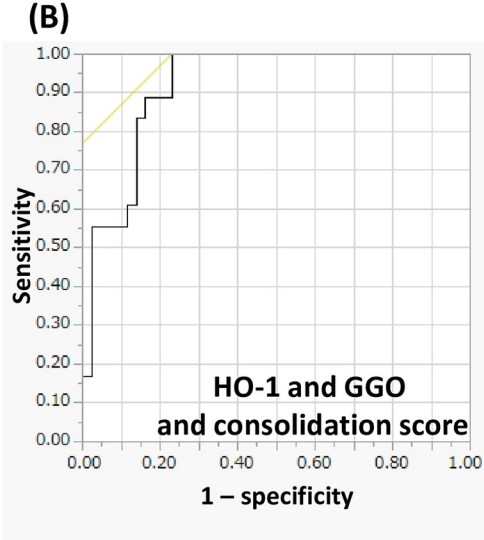

**(C)**

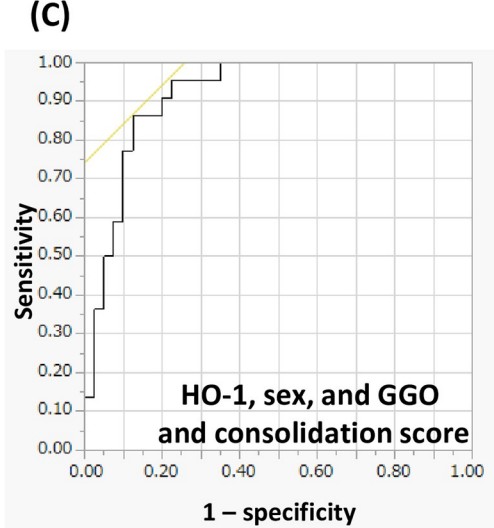

**(D)**

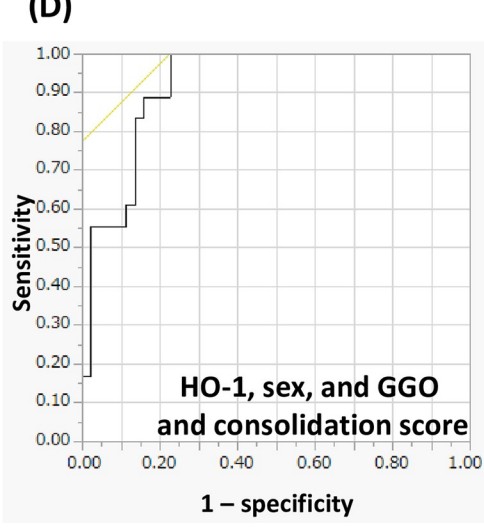

**Fig 3. Prediction of treatment outcomes using the combination of serum heme oxygenase (HO)-1 and ground glass opacity (GGO) and consolidation score.** (A) Serum HO-1 and GGO and consolidation score for predicting intensive care unit (ICU) admission: the area under the receiver operating characteristic curve (AUC): 0.915 (P < 0.001); (B) Serum HO-1 and GGO and consolidation score for predicting mechanical ventilation: AUC 0.919 (P < 0.001); (C) Serum HO-1, sex, and GGO and consolidation score for predicting ICU admission: AUC 0.916 (P < 0.001); (D) Serum HO-1, sex, and GGO and consolidation score for predicting mechanical ventilation: AUC 0.920 (P < 0.001). (A, C) Analysis of the prediction of ICU admission and mechanical ventilation support using composite parameters including serum HO-1, sex, and the GGO and consolidation score show that the combination has a higher the AUC than the AUCs of a single predictor (only HO-1) or the combination of two (serum HO-1 and GGO and consolidation score). (B, D) The composite parameters including serum HO-1, sex, and the GGO and consolidation score calculated for prediction of mechanical ventilation support shows a higher AUC than AUCs of a single predictor or a combination of two.

## Discussion

Patients with severe COVID-19 may exhibit features of systemic hyper-inflammation (cytokine storm) due to oxidative stress [10, 32]. Macrophages are critical contributors to immune responses in COVID-19 [9, 10], in which the interaction between M1 and M2 is closely correlated with disease progression in COVID-19 patients with ARDS [33]. Of note, the fact that SARS-CoV-2 distinctively hijacks M1 to proliferate and spread was demonstrated [15]. These findings suggest the importance of macrophage polarization in pathogenesis of COVID-19.

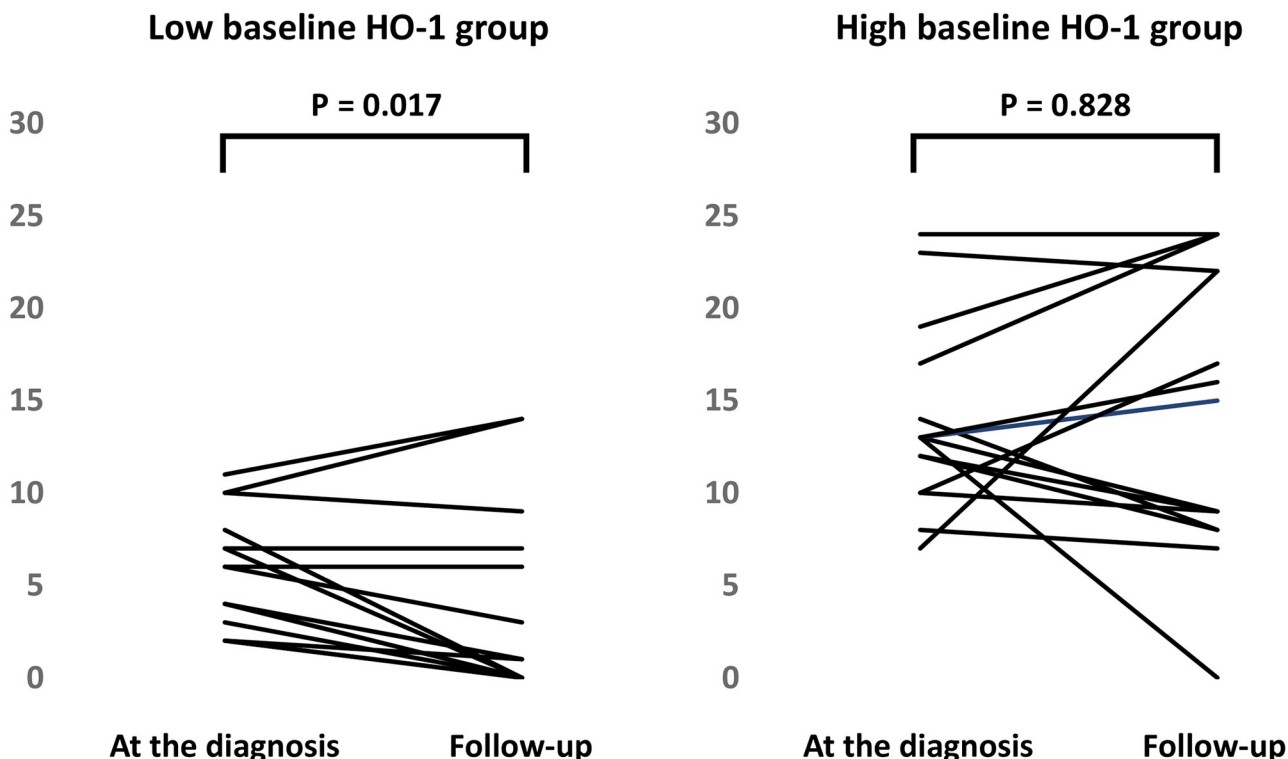

**Fig 4. Relationship between baseline serum heme oxygenase (HO)-1 and the serial change of ground glass opacity (GGO) and consolidation score.** We did follow up the HRCT scores in 31 patients and divided these cases into the low baseline HO-1 group (N = 16) and the high baseline HO-1 group (N = 15). We found that in the former group, the GGO and consolidation score were significantly decreased (A), while in the latter group, it was persistently high (B).

It has been reported that HO-1 is strongly and exclusively induced in M2 [14], and enhanced HO-1 gene expression, *HMOX1*, in COVID-19 [24]. The clinical significance of HO-1 in COVID-19 was not clear, so we investigated whether or not evaluating the degree of oxidative stress in COVID-19 by measuring serum HO-1 levels could be useful in predicting disease severity and treatment outcomes, and we compared these findings with the performance of another M2 biomarker, sCD163 [17–21]. In this respect, we were able to demonstrate that HO-1 can be a clinically useful and appropriate biomarker.

Increased HO-1 in alveolar macrophages was demonstrated in various pulmonary diseases such as ARDS and ILD, reflecting activated M2 against oxidative stress [25, 26]. Within the clinical course of ILD, an acute exacerbation (AE) can occur at any time and is associated with significant morbidity and mortality, and diffuse alveolar damage (DAD) is considered the histological hallmark of the acute phase of ARDS and AE of ILDs [28, 34]. The predominant histological findings of COVID-19 with respiratory failure were reported to be exudative DAD [35, 36]. Therefore, accurate detection of DAD and its severity assessment are very important for predicting the severity of COVID-19. HO-1 was previously reported to be significantly elevated in lung tissue and in bronchoalveolar lavage fluid taken from patients with ARDS [25]. In other previous research, measurement of serum HO-1 levels was reported to be useful to assess disease activity and in predicting a prognosis in patients with ARDS and AE of ILD [27, 28]. In addition to alveolar macrophages, since HO-1 is also abundantly produced by resident macrophages in reticuloendothelial tissues, a remarkable elevation of HO-1 in a cytokine storm can be caused by DAD-induced-ARDS in COVID-19 [33, 37]. It is a plausible

hypothesis that elevated serum HO-1 levels occur with the progression of COVID-19 severity. In this study, serum HO-1 increased with disease severity and was useful as a predictor of a prognosis on ROC analysis in COVID-19 cases. Furthermore, consistent with previously reported data, serum HO-1 was positively correlated with serum LDH, CRP, and the GGO and consolidation score as markers of cellular damage and alveolitis in the lung [30, 38]. Taken together, these findings suggest that serum HO-1 in COVID-19 may reflect disease severity or progression to DAD.

Although serum sCD163 is also considered an excellent biomarker of M2 in patients with COVID-19, it remains unclear if sCD163 is useful as severity assessment. In severe COVID-19, a finding concerning the importance of aberrant CD163[high] circulating monocytes and infiltrating the monocyte-driven M2 macrophages into lungs, leading to hyperinflammation of the airway, has been proven [39]. However, in a prospective study including 59 SARS-CoV-2 positive hospitalized patients, there were no differences of sCD163 between ICU and non-ICU infected patients [40]. In another study including 102 patients, sCD163 levels were higher in the ARDS group than in the non-ARDS group [20]. In addition, in our study, the results of the ROC analysis indicate that serum HO-1 is a better biomarker than sCD163 when evaluating disease severity associated with lung injury and predicting a prognosis. On the other hand, serum sCD163 showed a strong correlation with serum KL-6, which is widely used as a biomarker for lung fibrosis, unlike serum HO-1 (S1 Table) [41–43]. To date, the enhanced CD163 expression on alveolar/circulating M2 in ILD such as idiopathic pulmonary fibrosis (IPF) and systemic sclerosis has been demonstrated, respectively [44, 45]. Another finding noted that an accumulation of CD163-expressing macrophages in lungs is closely correlated with a prognosis of IPF patients [46]. Our findings suggest a close relationship in COVID-19 cases between sCD163 and KL-6, and this is supported by a recent finding reported by D. Wendisch *et al.* concerning the pivotal roles of recruited CD163-expressing M2 macrophages driven from circulating monocytes into lungs in the process of COVID-19-induced lung fibrosis [47]. Serum sCD163 may be an adequate biomarker reflecting lung fibrosis as a sequela of COVID-19. The present results of ROC analysis indicate that serum HO-1 is a better biomarker than sCD163 for evaluating of disease severity associated with lung injury and for predicting a prognosis in COVID-19 patients. Based on our results, we deem it necessary to verify the suitability of biomarkers predicting the severity of COVID-19. The serum HO-1 or sCD163 should therefore be investigated in greater depth and on a larger scale.

Composite approaches have been developed using peripheral blood biomarkers and physiological and radiographic measurements to provide more accurate prognostic information [48–50]. The composite scoring system, which is based on the arterial partial oxygen pressure (PaO$_2$)/fraction of inspiratory oxygen, serum LDH, KL-6, and the extent of abnormal CT findings, predicts 3-month mortality in AE of IPF patients [48]. In our previous analysis, we demonstrated that a composite scoring system including the Charlson Comorbidity Index score was useful for predicting survival in patients with ILD in both stable and AE state [49, 50]. In the present study, a composite parameter including serum HO-1, sex, and the GGO and consolidation score showed a higher AUC for predicting ICU admissions and mechanical ventilation than a single predictor (only HO-1) or a combination of the two (serum HO-1 and the GGO and consolidation score). Accordingly, evaluation of serum HO-1 is useful for predicting COVID-19 progression. It is possible to improve the prediction of disease progression by combining serum HO-1 with sex and the GGO and consolidation scores on HRCT findings.

Although serum HO-1 may prove to be a useful biomarker in patients with COVID-19, there are several limitations in the present study. Firstly, the enrolled patient number was small and from only a single institution. It is very important to validate the usefulness of the HO-1 assay in COVID-19 and also its integration into a multiparametric prognostic scores,

including clinical, radiological and functional data in a multi-center prospective study. Secondly, since more than a few patients had to be transferred to other hospitals after intensive care, the final outcomes of survivors and in-hospital deaths need to be accurately traced and recorded. Further verification will be necessary in order to determine the validity of serum HO-1 as a predictor of long-term survival. Thirdly, because we measured the serum HO-1 levels after the diagnosis of COVID-19 in the hospital, it was unclear if the real baseline serum HO-1 accurately reflected the levels at the start of infection. Also, we must evaluate whether or not the serial change of serum HO-1 could reflect disease progression from the initial stage of infection to DAD, though we previously reported on a case of AE of IPF that was triggered by COVID-19 in which continuous HO-1 measurements could be clinically useful in tracking severity and predicting disease prognosis [51]. Finally, based only on present data, whether or not the roles of HO-1 in COVID-19 is beneficial for patients could not be determined. Because HO-1 has a very high affinity for oxygen, which indicates that HO-1 will be saturated with oxygen in the blood over the physiological range of $PaO_2$, the consumption of oxygen by HO-1, nonetheless, is sustainable in severe hypoxemia caused by COVID-19. Namely, HO-1 is considered to have contradictory characteristics in COVID-19 [23, 52, 53].

## Conclusions

In conclusion, HO-1, reflecting activation of M2, could be a clinically useful biomarker of disease severity and a predictor of a prognosis in COVID-19 patients. Controlling M2 might be a therapeutic target for treating COVID-19 patients.

## Supporting information

**S1 Fig. Representative images of lung CT patterns of patients with COVID-19.** (A) Age 60 years, female. Ground glass opacity (GGO) and consolidation (GGO and consolidation score is 10 points). (B) Age 72 years, female. GGO and reticulation (GGO (without consolidation) and reticular fibrosis scores are 13 points and 1 point, respectively).
(TIF)

**S1 Table. The relationships between serum HO-1 or sCD163 and other clinical parameters.**
(TIF)

**S2 Table. Available data in the present study.**
(TIF)

## Author Contributions

**Conceptualization:** Yu Hara, Jun Tsukiji, Aya Yabe, Yoshika Onishi, Haruka Hirose, Masaki Yamamoto, Makoto Kudo, Takeshi Kaneko, Toshiaki Ebina.

**Data curation:** Yu Hara, Jun Tsukiji, Aya Yabe, Yoshika Onishi, Haruka Hirose, Masaki Yamamoto, Makoto Kudo, Takeshi Kaneko, Toshiaki Ebina.

**Formal analysis:** Yu Hara, Jun Tsukiji, Toshiaki Ebina.

**Funding acquisition:** Yu Hara, Jun Tsukiji, Takeshi Kaneko.

**Investigation:** Yu Hara, Jun Tsukiji, Aya Yabe, Yoshika Onishi, Haruka Hirose, Takeshi Kaneko, Toshiaki Ebina.

**Methodology:** Yu Hara, Jun Tsukiji, Aya Yabe, Yoshika Onishi, Haruka Hirose, Toshiaki Ebina.

**Project administration:** Yu Hara, Jun Tsukiji, Aya Yabe, Yoshika Onishi, Haruka Hirose, Toshiaki Ebina.

**Resources:** Yoshika Onishi, Masaki Yamamoto, Makoto Kudo, Toshiaki Ebina.

**Software:** Aya Yabe, Haruka Hirose.

**Supervision:** Yu Hara, Jun Tsukiji, Makoto Kudo, Takeshi Kaneko, Toshiaki Ebina.

**Validation:** Yu Hara, Jun Tsukiji, Aya Yabe, Haruka Hirose, Masaki Yamamoto, Makoto Kudo, Takeshi Kaneko, Toshiaki Ebina.

**Visualization:** Yu Hara, Jun Tsukiji.

**Writing – original draft:** Yu Hara, Jun Tsukiji.

**Writing – review & editing:** Yu Hara, Jun Tsukiji, Aya Yabe, Yoshika Onishi, Haruka Hirose, Masaki Yamamoto, Makoto Kudo, Takeshi Kaneko, Toshiaki Ebina.

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
