## [Decision Letter · Decision Letter 0]

13 Apr 2022

PONE-D-21-36509Heme oxygenase-1 as an important predictor of the severity of COVID-19PLOS ONE

Dear Dr. Tsukiji,

Thank you for submitting your manuscript to PLOS ONE. After careful consideration, we feel that it has merit but does not fully meet PLOS ONE’s publication criteria as it currently stands. Therefore, we invite you to submit a revised version of the manuscript that addresses the points raised during the review process.

We look forward to receiving your revised manuscript.

Kind regards,

Shih-Chang Hsu

Academic Editor

PLOS ONE

Journal Requirements:

Reviewers' comments:

Reviewer's Responses to Questions

**Comments to the Author**

1. Is the manuscript technically sound, and do the data support the conclusions?

Reviewer #1: Yes

Reviewer #2: Yes

2. Has the statistical analysis been performed appropriately and rigorously? 

Reviewer #1: Yes

Reviewer #2: I Don't Know

3. Have the authors made all data underlying the findings in their manuscript fully available?

Reviewer #1: Yes

Reviewer #2: Yes

4. Is the manuscript presented in an intelligible fashion and written in standard English?

Reviewer #1: Yes

Reviewer #2: Yes

5. Review Comments to the Author

Reviewer #1: 1. the authors should include patients inclusion and exclusion criteria in the methods section

2. Table 1, authors only reported male sex statistics, how about female sex?

3. Did author check lung function of the patients? it would be great to correlate HO-1 levels with lung function parameters.

Reviewer #2: in the methodology, the calculation of the scores is not clear

Are there any particularly interesting data in the deaths and what is the cause of their death

Figures are clear and well detailed

in this study you suggested that serum HO-1 in COVID-19 may reflect severity or progression to DAD. in the latter case it was rather necessary to study the serological kinetics of HO-1 for at least 6 months

in the methodology you did not specify the time of the serological analysis in relation to the start of the infection

the use of the composite prognostic score is a good practice in this study, but it was necessary to calculate these scores on admission and on leaving the hospital for cured patients and also to re-evaluate these scores during the follow-up of these patients

finally it is an interesting study but it remains to first validate the usefulness of the HO-1 assay in COVID-19 pneumonia and also its integration into a multiparametric prognostic score including clinical, radiological and functional data

6. PLOS authors have the option to publish the peer review history of their article (what does this mean?). If published, this will include your full peer review and any attached files.

Reviewer #1: No

Reviewer #2: **Yes: **KHADIJA AYED

---

## [Author Response · Author response to Decision Letter 0]

21 Jun 2022

Response to Reviewers

1. Concerning PLOS ONE’s style requirements 

The manuscript was revised according to style requirements. 

2. Concerning Informed Consent

Concerning informed consent for COVID-19 patients in the present study, the consent was not obtained from each patient because; 

(1) This retrospective analysis of existing data did not require any interaction with patients and did not intervene in their treatment.

(2) We had to avoid the opportunities of droplet, airborne and contact infection against SARS-CoV-2, which was mysterious pathogen at the time. 

Hence, in all patients, consent for participation of this retrospective study was obtained by disclosing a clinical study including the description of opt-out. We added this description (P11L198-P20L200). 

3. Concerning stating financial disclosure

We did not receive any funding for this study. Therefore, we state: “The authors received no specific funding for this work.” （P40L585-586）

4. Concerning Data Availability statement

The datasets used and/or analysed during the current study are available from the corresponding author on reasonable request.

5. Concerning Supporting Information files

The manuscript was revised according to Supporting Information guideline.

6. Concerning Additional References and Description of new contents

(1) We submitted the present our manuscript on the November 17th, 2021. Thereafter, some extremely critical findings associated with a pathogenesis of severe COVID-19 and with a content of our manuscript, have been published. For example, D. Wendisch et al. demonstrated pivotal roles of recruited CD163-expressing M2 alveolar macrophages into lungs from circulating monocytes on the process of COVID-19-induced lung fibrosis (Cell. 2021 Dec 22; 184(26): 6243–6261.e27.), which relates with our finding that serum sCD163 showed a strong correlation with serum KL-6, which is widely used as a biomarker for lung fibrosis. Therefore, we considered that our original manuscript should be partially re-touched to be precisely understood based on the newest scientific findings. 

So, we added some related references (reference number 39, 46, 47 and 51), and some sentences in ‘discussion’ (P23L308-311 and P24L321-327) to be understood more easily and precisely. 

(2) We changed disease name, COVID-19, correctly from ‘coronavirus infection disease’ to ‘coronavirus disease’ (P4L68 in ‘Keywords’, P5L75 in ‘Abbreviation lists’, and P6L95 in ‘Introduction’). We also correct all expression of “GGO score” in the original manuscript to “GGO and consolidation score” in the present manuscript.　Moreover, because ‘HMOX1’ is gene, we changed them to italic ‘HMOX1’(P8L130, P22L278). 

(3) Also, we changed a sentence to be understood precisely from “since HO-1 is abundantly produced by reticuloendothelial tissues, cytokine storm can be caused by DAD-induced-ARDS in COVID-19.” to “since HO-1 is also abundantly produced by resident macrophages in reticuloendothelial tissues, a remarkable elevation of HO-1 in cytokine storm can be caused by DAD-induced-ARDS in COVID-19” (P23L297-299).

(4) We reconfirmed the enrolled 64 patients in the present study. Among them, seven were outpatients. We have to correct the description in connection with this (P8L143, P9L161-162, P10L173, P12L206). 

(5) Blood samples to be measured were taken on admission basically. However, if no sufficient serum samples were left, the next samples which were taken another day were used to be measured. In the case of outpatient, a sample taken at the first visit to our hospital was used. We have to correct the description in connection with these points (P10L173). 

7. For Comments of Reviewers

Reviewer #1:1. The authors should include patients inclusion and exclusion criteria in the methods section. 

【Our reply】

Thank you for an indication. We clearly described that the inclusion/exclusion criteria of all COVID-19 patients in the present study were decided by PCR analysis, from nasopharyngeal and pharyngeal swab samples, on their admissions to the Yokohama City University Medical Center Hospital (P9L159-162).

Reviewer #1:2. Authors only reported male sex statistics, how about female sex?

【Our reply】

Thank you for your comment. We added the female information in Table 1.

Reviewer #1:3. Did author check lung function of the patients? It would be great to correlate HO-1 levels with lung function parameters.

【Our reply】

Thank you very much for bringing it to our attention. No pulmonary function tests were regrettably performed due to prevention of infection against SARS-CoV-2. We described this point clearly (P13L217-L218). 

Reviewer #2:1. In the methodology, the calculation of the scores is not clear

【Our reply】

Very important comment. We added the description for the calculation of the HRCT scores (P9L149-L157). For cases in which HRCT could be re-examined, the scoring was performed again and we examined the relationships between baseline serum HO-1 and the variation of HRCT scores (Figure 4)

Reviewer #2:2. Are there any particularly interesting data in the deaths and what is the cause of their death?

【Our reply】

That is just as you indicated.　However, six patients among severe cases died in clinical course in the hospital and no postmortem analysis were regrettably performed. Therefore, we could not get detail information about the cause of their death and there were no more informative data in the deaths. We described this point (P12L210-212).

Reviewer #2:3. In this study you suggested that serum HO-1 in COVID-19 may reflect severity or progression to DAD. in the latter case it was rather necessary to study the serological kinetics of HO-1 for at least 6 months

【Our reply】

Very important comment. In this study, we could not follow up the serum HO-1 (baseline only). However, we have previously experienced the case of AE of IPF triggered by COVID-19 that the continuous HO-1 measurement could be clinically useful for tracking severity and predicting disease prognosis (reference 51). Therefore, we added the sentences with future direction in the limitation section (P26L356-L362).

Reviewer #2:4. In the methodology you did not specify the time of the serological analysis in relation to the start of the infection.

【Our reply】

Very sorry, because we measured the serum HO-1 after the diagnosis of COVID-19 in our hospital, it was unclear the real baseline serum HO-1 accurately reflected the start of infection. In the future, it is important to evaluate the serological kinetics of serum HO-1 from initial stage of infection to DAD.

Reviewer #2:5. The use of the composite prognostic score is a good practice in this study, but it was necessary to calculate these scores on admission and on leaving the hospital for cured patients and also to re-evaluate these scores during the follow-up of these patients

【Our reply】

Very sorry, we measured the serum HO-1 at the diagnosis of COVID-19 only and could not re-evaluated the composite prognostic score including serum HO-1 and HRCT score. However, we could follow up the HRCT scores in 31 patients (46%) and calculated the GGO and consolidation score. We then divided these cases into high and low serum HO-1 groups and found that, in the former group, the GGO and consolidation score was significantly decreased (first GGO and consolidation score, 5 (2.3 - 7.8 points); second GGO and consolidation score, 0.5 (0 - 6.8 points) (P = 0.017)) (Fig. 4A), while, in the latter group, it was persistently high (first GGO and consolidation score, 13 (10 - 17 points); second GGO and consolidation score, 15 (8 - 22 points) (P = 0.828)) (Fig. 4B)　（P20L256-P21L267）. We also added the Figure 4 legend (P39L574-L579).

Reviewer #2:6. Finally it is an interesting study but it remains to first validate the usefulness of the HO-1 assay in COVID-19 pneumonia and also its integration into a multiparametric prognostic score including clinical, radiological and functional data.

【Our reply】

Very important. We added the above sentences in the limitation section (P26L348-L352). It is very important to validate the usefulness of the HO-1 assay in COVID-19 in a multi-center, prospective study.

---

## [Decision Letter · Decision Letter 1]

10 Aug 2022

Heme oxygenase-1 as an important predictor of the severity of COVID-19

PONE-D-21-36509R1

Dear Dr. Tsukiji,

We’re pleased to inform you that your manuscript has been judged scientifically suitable for publication and will be formally accepted for publication once it meets all outstanding technical requirements.

Kind regards,

Shih-Chang Hsu

Academic Editor

PLOS ONE

**Comments to the Author**

1. If the authors have adequately addressed your comments raised in a previous round of review and you feel that this manuscript is now acceptable for publication, you may indicate that here to bypass the “Comments to the Author” section, enter your conflict of interest statement in the “Confidential to Editor” section, and submit your "Accept" recommendation.

Reviewer #1: All comments have been addressed

Reviewer #2: All comments have been addressed

2. Is the manuscript technically sound, and do the data support the conclusions?

Reviewer #1: Yes

Reviewer #2: Yes

3. Has the statistical analysis been performed appropriately and rigorously? 

Reviewer #1: Yes

Reviewer #2: I Don't Know

4. Have the authors made all data underlying the findings in their manuscript fully available?

Reviewer #1: Yes

Reviewer #2: Yes

5. Is the manuscript presented in an intelligible fashion and written in standard English?

Reviewer #1: Yes

Reviewer #2: Yes

6. Review Comments to the Author

Reviewer #1: (No Response)

Reviewer #2: (No Response)

7. PLOS authors have the option to publish the peer review history of their article (what does this mean?). If published, this will include your full peer review and any attached files.

Reviewer #1: **Yes: **Md Khadem Ali

Reviewer #2: **Yes: **KHADIJA AYED

---

## [Editor Report · Acceptance letter]

15 Aug 2022

PONE-D-21-36509R1 

Heme Oxygenase-1 as an Important Predictor of the Severity of COVID-19 

Dear Dr. Tsukiji:

I'm pleased to inform you that your manuscript has been deemed suitable for publication in PLOS ONE. Congratulations! Your manuscript is now with our production department. 

Kind regards, 

on behalf of

Dr. Shih-Chang Hsu 

Academic Editor

PLOS ONE